# Simulating Meteorological and Water Wave Characteristics of Cyclone Shaheen

Mohsen Rahimian [1], Mostafa Beyramzadeh [1], Seyed Mostafa Siadatmousavi [1,*] and Mohammad Nabi Allahdadi [2]

[1] School of Civil Engineering, Iran University of Science & Technology, Narmak, Tehran 1684613114, Iran
[2] Department of Marine, Earth, and Atmospheric Sciences, North Carolina State University, Raleigh, NC 27695, USA
* Correspondence: siadatmousavi@iust.ac.ir

**Abstract:** The Bay of Bengal and Arabian Sea are annually exposed to severe tropical cyclones, which impose massive infrastructure damages and cause the loss of life in coastal regions. Cyclone Shaheen originally generated in the Bay of Bengal in 2021 and translated a rare east-to-west path toward the Arabian Sea. Although the cyclone's wind field can be obtained from reanalysis datasets such as ERA5 (fifth generation European Centre for Medium-Range Weather Forecasts), the wind speed cannot be reproduced with realistic details in the regions close to the center of cyclone due to spatial resolution. In this study, to address this problem, the high-resolution advanced Weather Research and Forecasting (WRF) model is used for simulation of Shaheen's wind field. As a critical part of the study, the sensitivity of the results to the planetary boundary layer (PBL) parameterization in terms of the track, intensity, strength and structure of the Cyclone Shaheen is investigated. Five experiments are considered with five PBL schemes: Yonsei University (YSU); Mellor–Yamada–Janjić (MYJ); Mellor–Yamada–Nakanishi–Niino level 2.5 (MYNN); Asymmetric Convective Model version 2 (ACM2); Quasi-Normal Scale Elimination (QNSE). The track, intensity, and strength of the experiments are compared with the wind fields obtained from the Joint Typhoon Warning Centre (JTWC) dataset. The results imply the high dependency of the track, intensity, and strength of the cyclone to the PBL parameterization. Simulated tracks with non-local PBL schemes (YSU and ACM2) outperformed those of the local PBL schemes (MYJ, MYNN, and QNSE), especially during the rapid intensification phase of Shaheen before landfall. The YSU produced highly intensified storm, while the ACM2 results are in better agreement with the JTWC data. The most accurate track was obtained from the ERA5 data; however, this dataset overestimated the spatial size and underestimated the wind speed. The WRF model using either YSU or ACM2 overestimated the wind speed compared to that of the altimeter data. The YSU and ACM2 schemes were able to reproduce the observed increase in wind speed and pressure drop at in situ stations. The wind data from EAR5 and cyclone parametric model were applied to the SWAN model to simulate the wave regime in the Arabian Sea during the time that Shaheen was translating across the region. Janssen formulation for wind input and whitecapping dissipation source terms in combination with both ERA5 and hybrid wind were used and the minimum combined error in the prediction of significant wave height ($H_s$) and zero up-crossing wave period ($T_z$) was examined. The maximum significant wave height for hybrid wind is higher than that of ERA5, while the cyclone development was successfully inferred from the wave field of the hybrid data.

**Keywords:** Arabian Sea; Cyclone Shaheen; ERA5; WRF; YSU; ACM2; SWAN; JANSSEN

## 1. Introduction

Tropical cyclones (hereinafter, TC) are the severe meteorological phenomena leading to substantial life loss and property damage. High wind speeds and large wave heights, severe rainfalls, floods, and storm surges are the direct consequence of TCs, which threat coastal communities. Severe winds caused by a cyclone are responsible for most of the

oceanic-related damages (including waves and storm surges) during a TC, while the intense wind itself is a significant threat. Hence, wind field simulation during the TCs is crucial to assist authorities and scientists in minimizing the associated damages.

The meteorological parameters of a TC including wind fields, sea level pressure, and maximum sustained wind speed are mostly available through three main sources: (1) Wind from atmospheric models such as reanalysis wind data from European Centre for Medium-Range Weather Forecasts (ECMWF) and Climate Forecast System Reanalysis (CFSR), which have been applied widely for wave simulation during TCs [1]. Hodges, et al. [2] compared the track and intensity of TCs extracted from reanalysis wind data with the IBTrACS (International Best Track Archive for Climate Stewardship) database. They showed that the tracks obtained from reanalysis wind data are sufficiently accurate, while the wind speed and pressure are under- and overestimated, respectively. In addition to the wind field, some imperfections in humidity, temperature, precipitation, and evaporation by reanalysis data have been described in many previous studies. Virman, et al. [3] compared the meteorological parameters from ERA5 and ERA-Interim against radiosonde observations over tropical oceans. They concluded that temperature and relative humidity by ERA5 and ERA-Interim deviate from the observations at many atmospheric levels; however, ERA-Interim is in more agreement with the observations than ERA5 is from the low- to mid-troposphere. Jiao, et al. [4] evaluated the temporal and spatial performance of ERA5 precipitation from 1979 to 2018 against observations in China. Although the ERA5 reproduced temporal and spatial patterns of measured data with high accuracy, precipitation was overestimated in the Chinese mainland during summer. The low correlation (~0.57) of precipitation water vapor by ERA5 and radiometer data in a short-duration assessment was observed, while for longer durations, the consistency of ERA5 precipitation water vapor with radiometric measured data was improved [5]. (2) Parametric wind models: these models representing the TC wind field based on gradient wind balance can substantially resolve the inaccuracy problem of atmospheric models in reproducing wind speed around the TC center [6,7]. (3) TC simulation uses mesoscale metrological models such as Weather Research Forecast (WRF). Several physics schemes for microphysics, the surface layer, and the planetary boundary layer (PBL) and convections have been implemented in the WRF model. These physics play crucial role in TC development. It is a challenging task to find the optimum configuration of a physics scheme for each specific TC. Mazyak and Shafieefar [8] used data from the QuikSCAT Scatterometer and coastal synoptic stations to skill assess the WRF model, parametric methods, and reanalysis data during the dominance of Cyclone Gonu in the Gulf of Oman. Their results indicated that the WRF model outperformed the parametric methods and reanalysis wind data in reproducing in situ observations. Alimohammadi and Malakooti [9] evaluated several domain configurations and physical parameterization settings for PBL, cumulous convection (CC), and ocean–atmosphere surface flux to improve the prediction of Cyclone Gonu in the Gulf of Oman. Their results proved that it is not possible to apply one specific PBL in the WRF model for all cyclones and determine their track and intensity accurately. Alimohammadi, et al. [10] scrutinized the role of momentum and enthalpy roughness length for the Cyclone Gonu simulation using the WRF model simulations. The track and intensity were well estimated using the formulation presented by Donelan, et al. [11] for the momentum roughness length, and Large and Pond [12] parameterization for enthalpy roughness length. The COAWST (Coupled Ocean–Atmosphere–Weather–Sediment Transport) framework [13], which uses SWAN (Simulating WAves Nearshore) and WRF as wave and atmosphere counterparts, was applied by Alimohammadi, Malakooti and Rahbani [10] to exchange data between wind and wave models for simulating the Cyclone Gonu wind field. The dependency of momentum roughness length on the wave age and peak wave length resulted in the overestimation of the roughness length. Nadimpalli, et al. [14] used 62 forecast setups and compared the performances of HWRF (Hurricane Weather Research and Forecasting) and WRF models for simulating 10 TCs during 2013–2017 in the Bay of Bengal. The predicted tracks were identical with both models for short period forecasts (shorter than 30 h), while

HWRF was more reliable for longer period forecasts. Moreover, the intensity of TC estimated by the HWRF model was more accurate than the one produced by the WRF model. Chutia, et al. [15] conducted sensitivity analysis on microphysics in WRF model to simulate a sever cyclonic storm. They proved that a three-class single moment with ice, excluding graupel or hail, was sufficiently accurate in reproducing the important features of a cyclone such as wind speed and sea level pressure.

This study aims to assess the performance of different PBL schemes within the context of a high-resolution atmospheric model, as well as the associated wind-induced waves during the presence of Cyclone Shaheen in the Arabian Sea. The WRF model is used to simulate the atmospheric parameters, and the SWAN model is employed to determine the resulting wave regime. In situ measurements, the study area, the modeling system, and the modeling approach with WRF and SWAN models are described in Section 2. Section 3 includes the skill assessment of models, followed by conclusions in Section 4.

## 2. Material and Methods

The track of cyclones during 1977–2016 over the Arabian Sea were extracted from JTWC (Joint Typhoon Warning Center) by Mazyak and Shafieefar [8]. In their research, three main routes were identified: (1) those moving to the north and tending to approach the Gulf of Oman and the Persian Gulf, which mainly impact the northern coast of Oman and the southern coast of Iran; (2) those moving to the north with a track that is deviated mostly to the east which impact the coastal regions of Pakistan and India; (3) those moving from the east to the west in the Arabian Sea. It is noteworthy to mention that the track types (1) and (2) are more frequent than track type (3) is in the Arabian Sea.

Two consecutive tropical systems were dominant in the Bay of Bengal and Arabian Sea from late September to early October 2021. The Gulab tropical depression was formed at 1800 UTC on 23 September. It passed the west of the Bay of Bengal and intensified as a tropical storm at 0000 UTC on 25 September, with a maximum sustained wind speed ($V_{max}$) of 35 knots. At this time, the northern Arabian Sea was still under the effect of summer monsoon, dominating the wind regime of the region from July to September [16]. The intensification phase continued up to 1800 UTC on 25 September when $V_{max}$ reached 40 knots and the minimum central pressure ($P_c$) dropped to 996 hPa. The Gulab experienced its lowest $P_c$ of 991 hPa at 1200 UTC on 26 September. From late 26 to early 27 September, it made landfall and weakened while it was over the terrain of India. Cyclone Shaheen was formed from the remnant of Gulab as a tropical depression. It entered the Arabian Sea at 0000 UTC on 30 September, with $V_{max}$ = 30 knots. It intensified to a category 1 TC, with $V_{max}$ = 70 knots and $P_c$ = 976 hPa at 1800 UTC, on 2 October (see Figure 1). Cyclone Shaheen made landfall on the northern coast of Oman as a category 1 hurricane at 0600 UTC on 3 October.

The evolution of cyclone Shaheen from 0500 UTC on 30 September to 3 October 2021 captured by Sentinel-3 satellite is shown in Figure 2. The large extent of Cyclone Shaheen as it entered the Arabia Sea is shown in panel (a). As it entered the Gulf of Oman (Figure 2c), it became smaller in size, most likely due to frictional effects imposed by the surrounding lands in the north and south.

At 0000 UTC on 30 September 2021, it scattered into broken, low and medium clouds with embedded intense-to-very-intense convection, which formed over the north-east and eastern central Arabian Sea. Afterward, the system was a severe cyclonic storm over the north-east Arabian Sea that moved west and north-westwards. In the landfall stage, the system moved west–south-westwards and weakened into a deep depression, associated with broken, low/medium clouds with embedded moderate-to-intense convection over Oman and the Gulf of Oman.

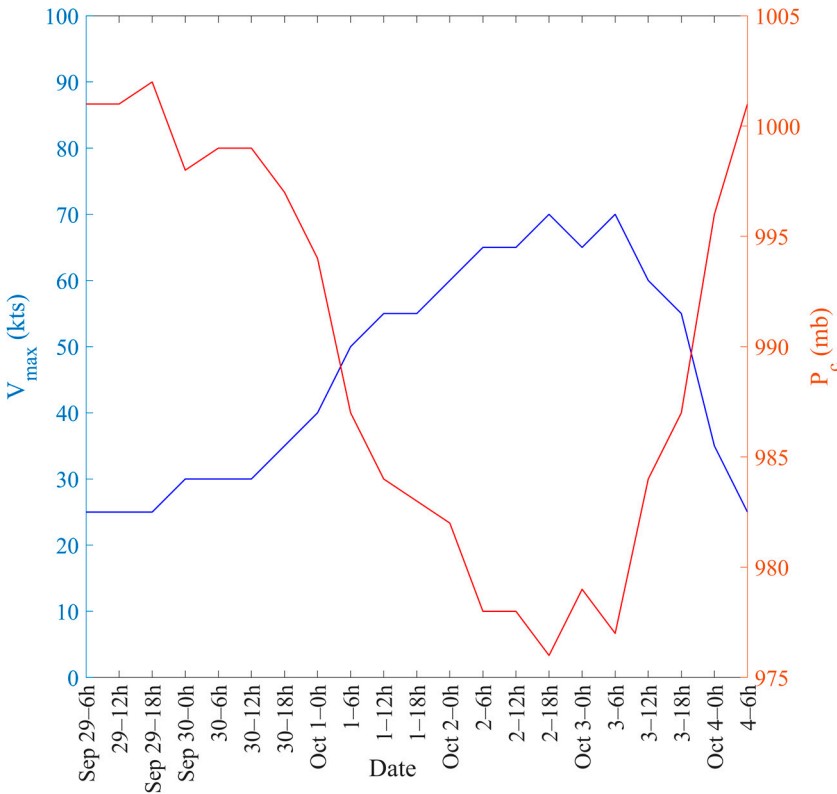

**Figure 1.** Extracted $V_{max}$ (blue line) and $P_c$ (red line) from JTWC during Cyclone Shaheen in the Arabian Sea.

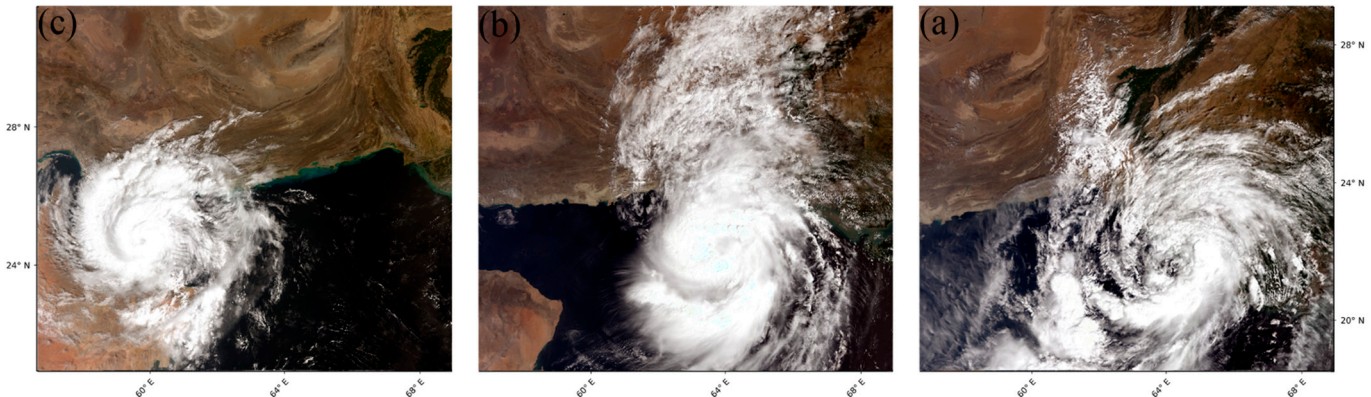

**Figure 2.** Cyclone Shaheen at 0500 UTC (**a**) on 30 October (**right** panel), (**b**) on 1 September (**middle** panel), and (**c**) on 3 September 2021 (**left** panel) as observed by Sentinel-3.

The unique track of Cyclone Shaheen and the damages over the northern coast of Oman caused by waves, storm surge, coastal inundation, and flash flooding are great motivations to study meteorological and water wave characteristics of the Gulf of Oman during the passage of this cyclone.

In present study, measurements from weather stations from 30 September to 4 October 2021 were used for the assessment of the WRF model. The location of these stations are shown in Figure 3. Additionally, the hourly ERA5 reanalysis data have been compared with the model results. Surface station data were obtained from two airport synoptic stations and a buoy station located in the Gulf of Oman. Data from synoptic stations were obtained from the National Climate Data Center (NCDC 2010). Hourly data from Chabahar wave recorder were provided by the Port and Maritime Organization of Iran. Wind speed derived from altimeters is another form of data for model assessment. An

altimeter dataset provided by the Australian Ocean Data Network (AODN) and retrieved from https://portal.aodn.org.au on 28 February 2019, was used. It included wind velocity along 14 satellite tracks over the Arabian Sea, as shown by Ribal and Young [17].

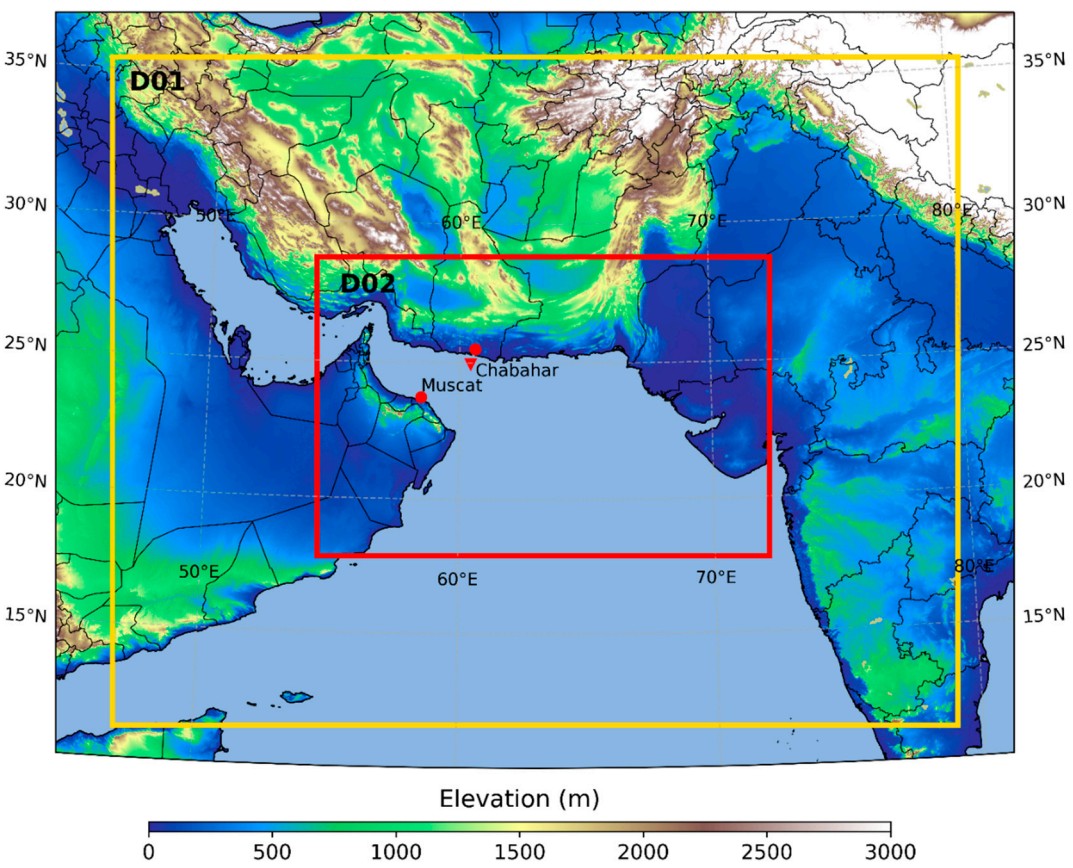

**Figure 3.** Domain configuration for the simulations of Cyclone Shaheen using WRF model with ARW core. Synoptic stations (red circle) and wave buoy (triangle) are also shown. The outer and inner grids are D01 and D02, respectively. The terrain height data were derived from the GEBCO 2022 dataset.

### 2.1. Wind Model

The fully compressible, non-hydrostatic mesoscale Advanced Research WRF (ARW) model is a state-of-the art numerical weather prediction (NWP) model designed to serve and support both atmospheric research and operational forecasting needs [18,19]. It was developed as a collaborative effort among several multi-agencies such as the National Oceanic and Atmospheric Administration (NOAA) and the National Center for Environmental Prediction (NCEP) and various universities. The WRF model is considered to be a suitable choice for the simulation of atmospheric features due to its accuracy, small truncation errors, resolving scale interactions, access to high-resolution information, and detailed physical parameterizations [20]. The WRF-ARW core is based on fully compressible, non-hydrostatic Euler equations with the hybrid sigma-pressure vertical coordinate system. The horizontal grid is the Arakawa-C grid. The second- or third-order Runge–Kutta scheme is used in the model for the time-split integration scheme. The spatial discretization is based on second-to sixth-order schemes. More details can be found in [19].

The WRF-ARW model has been used by many researchers to study the effect of model physics on the prediction of cyclone characteristics over the north Indian ocean including the Arabian Sea and the Bay of Bengal [9,21–26]. In this study, WRF version 4.3 is configured with two-way interactive nested domains to estimate the pressure and wind field of Cyclone Shaheen. The outer domain covers a larger region (46.25–79.75° E; 11.45–35.44° N; D01) with 9 km horizontal resolution and 380 × 300 grid points, while the inner domain has a 3 km horizontal resolution with 610 × 403 grid points (54.29–72.24° E;

17.85–28.71° N; D02). The model comprises the Arabian Sea and the Gulf of Oman and have the Lambert Mercator map projection. The finer domain (D02) fully covers the spatial extent of Cyclone Shaheen during the whole simulation period. The domain configuration is presented in Figure 3. The model includes 50 vertical atmospheric layers, up to 50 hPa pressure, and on 4 soil layers. The simulation time step was 36 s for D01 and 12 s for D02.

The initial and boundary conditions were obtained from the NCEP-FNL operational global analysis, for which forecast data are available at a 0.25° × 0.25° (~27 km) horizontal resolution, with time varying boundary conditions updated every 6 h. The topography data used for the outer and inner domains were obtained from the GMTED2010 database at 30" resolution and the land use was from modified IGBP 21-category, 15 arc seconds MODIS LULC database. The topography within the domain is shown in Figure 3. Sea surface temperature (SST) is known as a key factor in cyclone dynamics, so this variable was taken as a time-varying boundary condition and was updated every 6 h. The SST data over all the domains were obtained from the GFS/FNL database. Simulations were initiated at 0000 UTC on 29 September 2021 with lateral boundary conditions and were carried for out up to 1200 UTC on 4 October 2021. The predicted track and intensity were compared with the JTWC observed cyclone track and retrieval cyclone intensity data.

An overview of the applied model physics parameterizations used in the simulations are presented in Table 1. Physical parameterizations schemes such as CC and cloud microphysics and PBL have a crucial role in the creation, evolution, and intensification of cyclones [27–30]. The PBL scheme is responsible for variations in the vertical fluxes due to eddy transports within the boundary layer and stable layer. PBL parameterization describes the vertical profiles of temperature, moisture and horizontal momentum and influences the variations of turbulence, winds, and other state variables in the lower atmosphere. Additionally, it is an important factor in cyclone generation and development because it affects heat, moisture, and momentum transports [31]. Hence, a set of five numerical experiments are performed with different PBL parameterizations to analyze the estimations of intensity and trajectory of Cyclone Shaheen (Table 1). To determine the sensitivity of the simulation to PBL, the following schemes are considered: The Yonsei University (YSU, [32]), the Mellor–Yamada–Janjić (MYJ, [33]), Mellor–Yamada Nakanishi and Niino (MYNN, [34]), Asymmetric Convective Model (ACM2, [35]) and Quasi-Normal Scale Elimination ones (QNSE, [36]). Although it is desirable to evaluate each combination of physical parameters in choosing the best physical ensemble, due to the lack of computational resources, only the effect of the boundary layer parameter was considered.

The YSU scheme is a first-order "non-local *K*" diffusion scheme, which incorporates an explicit entrainment layer and parabolic *K* profile in an unstable mixed layer [32]. It uses a similarity relationship for the heat flux and the drag formulation following the Charnock [37] approach. The MYJ scheme is a "local *K*", 1.5 order, turbulence closure scheme [33,38]. It determines eddy diffusion coefficients by solving a prognostic equation for Turbulent Kinetic Energy (TKE) using local vertical mixing for both convective and stable PBL. The MYNN scheme uses a level 2.5 turbulence closure based on TKE with local vertical mixing [34]. It can be configured to run at 3.0 level closure as well. The height of the PBL is determined by TKE production. In order to improve the insufficient growth of the convective boundary layer in the MYJ scheme, the MYNN scheme is adjusted with large eddy simulations (LES) [39]. The ACM2 scheme is a non-local scheme in an unstable condition, which includes a first-order eddy diffusion scheme [35]. The ACM2 scheme considers the non-local fluxes explicitly through a transient term [40]. The QNSE scheme uses a new spectral quasi-Gaussian spectral closure model for a stably stratified regions to account for the wave phenomena within the stable boundary layer [36]. It is a 1.5 order, local closure scheme and has a TKE prediction option based on vertical mixing in the boundary layer and free atmosphere. The MYJ, MYNN, and QNSE PBL schemes are classified as TKE closure schemes that are different in how they resolve the mixing length. The other physics options were chosen based on earlier studies conducted for cyclones over the northern Indian Ocean [22,24,41]. The physics options selected include the Thompson microphysics

scheme [42], Dudhia shortwave radiation [43], Rapid Radiative Transfer Model (RRTM) longwave radiation [44], Grell–Devenyi ensemble cumulus scheme [45], and the unified Noah land surface model [46]. The model physics options were the same for all the domains, except that no cumulus parameterization was included for the inner domain.

**Table 1.** Main WRF physical options selected for the study.

| Component | Scheme Adopted |
| --- | --- |
| Microphysics scheme | Thompson |
| Cumulus physics scheme | Grell–Devenyi Ensemble |
| Shortwave radiation scheme | Dudhia |
| Longwave radiation scheme | RRTM |
| PBL scheme | Yonsei University (YSU) |
| | Mellor–Yamada–Janjić (MYJ) |
| | Mellor–Yamada–Nakanishi–Niino level 2.5 (MYNN) |
| | Asymmetric Convective Model version 2 (ACM2) |
| | Quasi-Normal Scale Elimination (QNSE) |
| Surface layer | Revised MM5 scheme [47] in combination with YSU |
| | Eta Similarity Scheme [33] with MYJ and MYNN |
| | Pleim–Xiu Scheme (Pleim [48]) with ACM2 |
| | QNSE Scheme [36] |
| Land surface model | Unified Noah Land Surface Model |

The parametric wind models for cyclones successfully reproduced the wind field around the center of cyclones, while reanalysis wind data (in this study ERA5) are more accurate at regions away from the center of cyclones; hence, in many previous research studies, the quality of the reanalysis wind data was enhanced by blending them with the parametric methods [49–52]. Following Mazyak and Shafieefar [8,53], Holland parametric method was applied to reproduce the wind field around the center of Cyclone Shaheen [54]. The blend of reanalysis ERA5 and Holland parametric method (hereinafter, hybrid) was used to force the wave model.

*2.2. Wave Model*

In this study, the phase-averaged spectral wave model, SWAN, was applied to simulate wave field in the Arabian Sea during the presence of Cyclone Shaheen in the region. This third generation wave model computes the evolution of wave action density using the action balance equation [55]:

$$\frac{DN}{Dt} = \frac{S_{tot}}{\sigma} \tag{1}$$

where $N$ is the wave action equal to $\frac{E}{\sigma}$, in which $E$ is the energy density function and $\sigma$ is the angular frequency. The left side of Equation (1) includes the temporal and spatial variations in wave action density, refraction, and radian shifting due to depth and current effects. The right hand side of Equation (1) represents source and sink terms, including the wind input term ($S_{in}$), whitecapping dissipation ($S_{wc}$), and non-linear transfer of wave energy by quadruplet that is dominant in deep water. Other mechanisms such as depth-induced wave breaking, bottom friction, and triad interaction are dominant in shallow coastal regions. Due to dependency of whitecapping dissipation with wind input, this term and its counterpart wind input are formulated together as different packages in wave models.

The default formulation for $S_{in}$ and $S_{wc}$ terms in SWAN model was suggested by Komen, et al. [56] (hereinafter, Komen). This method is the simplest formulation for wind input and whitecapping dissipation terms in the SWAN model. Its wind input is based

on Snyder, et al. [57] and depends on the inverse wave age ($\frac{u_*}{C_p}$, where $u_*$ is the friction velocity and $C_p$ is the peak phase speed). $S_{wc}$ for Komen depends on the average wave number and steepness:

$$S_{wc}(\sigma,\theta) = -C_{ds}\left(\frac{k}{\widetilde{k}}\right)^2\left(\frac{\widetilde{s}}{\widetilde{s}_{PM}}\right)^4\widetilde{\sigma}E(\sigma,\theta) \tag{2}$$

In Equation (2), $C_{ds}$ is a coefficient with a default value $2.36 \times 10^{-5}$. $\sigma$, $k$, $\theta$, $\widetilde{\sigma}$ and $\widetilde{k}$ denote radian frequency, wave number, wave direction, mean frequency, and mean wave number, respectively. Additionally, $\widetilde{s}$ is the overall wave spectral steepness, and $\widetilde{s}_{PM}$ represents the steepness of the Pierson–Moskowitz spectrum [58]. In present study, Komen was used with suggested default tuning values in the SWAN model.

WAM-Cycle4 (hereinafter, Janssen) is an alternative to Komen formulation in the SWAN model. An iterative process is needed to include the feedback of wave-induced shear on the wind profile. The whitecapping dissipation for this package is presented in Equation (3). It includes $C_{ds}$ parameter to tune the intensity of dissipation, and two weighted coefficients, $1 - \delta$ and $\delta$, to change its frequency distribution [59]. Moreover, $E_{tot}$, $\widetilde{k}$, and $\widetilde{\sigma}$ are the total wave energy, mean wave number, and mean angular frequency, respectively. In the default mode of Janssen, $C_{ds} = 4.5$ and $\delta = 0.5$ (hereinafter, Janssen-Default). Beyramzadeh, et al. [60] skill assessed the WAVEWATCH-III model against in situ and altimeter data with using the Janssen method. Lower $C_{ds}$ and $\delta$ values (3.5 and 0.3, respectively) were proposed as an optimum tuning values in the Gulf of Oman (hereinafter, Janssen-BSD2021).

$$S_{wc}(\sigma,\theta) = -C_{ds}\left[(1-\delta)+\delta\frac{k}{\widetilde{k}}\right]E_{tot}^2\widetilde{k}^3\widetilde{\sigma}E(\sigma,\theta) \tag{3}$$

The dependence of whitecapping dissipation on the mean wave number and steepness leads to overestimation of energy at high-frequency tail of the spectrum in the presence of swell wave. This spurious behavior of the model could result in peak and mean period underestimation. Alves and Banner [61] developed the whitecapping dissipation term based on the level of the saturation threshold, but some multiplication factors keep the dependency of method to mean steepness and mean wave number. van der Westhuysen, et al. [62] (hereinafter, Westhuysen) revisited this method to resolve peak and mean period underestimation by the SWAN model under mixed sea–swell conditions. Three modifications were applied: (1) The suggested threshold value in [61] is too high and should be decreased; (2) the whitecapping expression was revisited to scale it to be similar to the wind input term proposed by Yan [63]. This scaling guarantees the $f^{-4}$ spectral form in the equilibrium range. (3) We retained the dependency of the whitecapping dissipation term on the frequency of locally breaking waves by removing items, which include mean steepness and mean wave number. The Westhuysen in the SWAN model successfully reproduced wave field during the dominance of Cyclones Gonu, Ashobaa, and Phet in the Arabian Sea [25]. The whitecapping dissipation ($S_{wc}$) term for Westhuysen is as follows:

$$S_{wc}(\sigma,\theta) = -C_{ds}\left[\frac{B(k)}{B_r}\right]^{\frac{p}{2}}(\tanh(kd))^{\frac{2-p_0}{4}}\sqrt{gk}E(\sigma,\theta) \tag{4}$$

In Equation (4), $C_{ds}$ is set to $5 \times 10^{-5}$ as a default value in the SWAN model. $B(k)$ and $B_r = 1.75 \times 10^{-3}$ denote spectral saturation and threshold saturation levels, respectively. Additionally, $d$ represents water depth, $p_0$ depends on inverse wave age $\frac{u_*}{C_p}$, and $p$ is formulated as a function of $p_0$, $B(k)$ and $B_r$. The default parameters were considered in present study (hereinafter, Westhuysen-Default).

In the most recent whitecapping dissipation formulation added to SWAN (*ST6*), the wind input term includes the effect of opposing wind as negative wind input. Similar to

wind input term in the Komen method, wind speed is scaled with $u_*$ (but with $32u_*$ instead of $28u_*$) [60,64–68]. The dissipation in *ST6* is presented as follows:

$$S_{wc}(k, \theta) = [T_1(k, \theta) + T_2(k, \theta)]N(k, \theta) \tag{5}$$

$$T_1(k) = a_1 A(k) \frac{\sigma}{2\pi} \left[ \frac{\Delta(k)}{\widetilde{N}(k)} \right]^{p_1}, \quad T_2(k) = a_2 \int_0^k A(k) \frac{c_g}{2\pi} \left[ \frac{\Delta(k)}{\widetilde{N}(k)} \right]^{p_2} dk \tag{6}$$

where $\Delta(k) = N(k) - N_T(k)$, $N_T$ is wave action threshold level, $c_g$ is wave group velocity, $a_1$ and $a_2$ are tuning parameters, $\widetilde{N}(k)$ is wave generic action density spectrum, and $A(k)$ is directional narrowness function, and $p_1 = p_2 = 4$. The $T_2$ term was added to general dissipation term $T_1$ to include the effect of long waves on the dissipation of short waves [69,70]. In this study, two sets of tuning values were considered: (1) $a_1 = 2.8 \times 10^{-6}$, $a_2 = 3.5 \times 10^{-5}$, $p_1 = p_2 = 4$, and a swell dissipation according to [70] with constants 0.00025 and 0.04 for negative wind input during opposing wind (hereinafter, ST6-Default); (2) $a_1 = 3.74 \times 10^{-7}$, $a_2 = 5.24 \times 10^{-6}$, $p_1 = p_2 = 4$, and a swell dissipation with same method as the former setup with constants 0.0032 and 0.09 for negative wind input during opposing wind (hereinafter, ST6-BSD2021) proposed by Beyramzadeh, Siadatmousavi and Derkani [60] as an optimum tuning values for ST6 package in the Gulf of Oman. The default parameters of ST6 formulation have recently been successfully used for the simulation of low-energy waves in the Gulf of Mexico [71].

The computational grid in the SWAN model covers 55–71° E and 21–27.4° N using a rectangular grid with $0.032° \times 0.032°$ resolution. Following [72], 30 frequencies were considered with a 10% incremental rate geometrically distributed in the range of 0.04–0.63 Hz. Moreover, 36 directions with $10°$ resolution were applied. The computational time step was set to 120 s in the model. The quadruplet interactions term was employed using the Discrete Interaction Approximation (DIA) method [73]. The depth-induced wave breaking was considered according to [74]. The bottom friction was included in the model using the JONSWAP formulation [75] with the constant of $-0.038$ $m^2s^{-2}$, according to [76].

Wind field, bathymetry, and boundary conditions are crucial for wave modelling. Simulated wind field using WRF model and ERA5 wind data were two available wind sources used to simulate the wave field in this study. These two wind sources were skill assessed in Section 3 against all available in situ data to select most appropriate wind source for wave simulations using Komen, Janssen, Westhuysen, and ST6 physics packages in SWAN. The hybrid wind field is another option considered for wave simulation, which was used with only Janssen-Default and ST6-Default in SWAN. Gridded bathymetry data were extracted from GEBCO (The General Bathymetric Chart of the Oceans), which has been available with a fine resolution of 0.004° since 2019. The Arabian Sea experiences southerly swells from the Indian Ocean; hence, reliable boundary condition data are necessary for an accurate wave simulation. The hourly boundary conditions were extracted in the form of directional wave spectra on the west (55° E) and south (21° N) boundaries from a global wave simulation following [60].

### 2.3. Statistical Indices

Three statistical indices including the mean bias error (*MBE*), the root mean square error (*RMSE*), and the index of agreement (*d*) [77] were used to skill assess the WRF and SWAN models against the measurements. These error indices are as follows:

$$MBE = \frac{\sum(M_i - O_i)}{N} \tag{7}$$

$$RMSE = \sqrt{\frac{1}{N} \sum(M_i - O_i)^2} \tag{8}$$

$$d = 1 - \frac{(M_i - O_i)^2}{\sum(|M_i - \overline{O}| + |O_i - \overline{O}|)^2} \tag{9}$$

in which $M_i$ and $O_i$ are modeled and observed quantities, respectively, while $N$ is the total number of observations. All indicators were calculated using hourly data. Note that $d$ is a dimensionless index, which quantifies the agreement between the two series of data; a value of $d$ index that is larger than 0.5 indicates a good performance of the model. Rahimian, et al. [78] evaluated the performance of WRF and WAVEWATCH-III models using these indices during the dominance of a meteotsunami in the Persian Gulf in 2017.

## 3. Results

### 3.1. Skill Assessment of WRF Model

Determinations of the cyclone track and the position of landfall, where high waves and storm surges threat coastal regions, are important outputs of an atmospheric model. Figure 4 shows the tracks of Cyclone Shaheen obtained from the WRF model using different PBL schemes, ERA5 reanalysis data, and the JTWC database. The WRF track resulting from the YSU and ACM2 schemes for PBL was in good agreement with the JTWC track, while other PBL schemes could not reproduce Cyclone Shaheen's track accurately. The initial stage of Cyclone Shaheen in the Arabian Sea predicted by the YSU and ACM2 schemes was almost identical and was associated with lower latitudes than that of JTWC. The track of Cyclone Shaheen simulated by ACM2 mimicked the JTWC data with a final landfall including a downward shift, but the cyclone predicted by YSU moved at lower latitudes than the real path of the Cyclone before its landfall. The predicted landfalls by YSU and ACM2 were approximately 6 h earlier and 12 h later than the observed one, respectively. Although the MYJ scheme predicted the track fairly well until 0600 UTC on 3 October, the cyclone eventually dissipated over the Arabian Sea and did not make landfall close to the Oman coast. The MYNN and QNSE schemes were not successful at reproducing the track at all. Similar to MYJ, the cyclone obtained by these schemes dissipated in the middle of the Arabian Sea early on 3 October. Note that the track predicted by ERA5 reanalysis data was in good agreement with the observations.

The temporal variation in the error in tracks predicted by YSU, ACM2, and MYJ formulations for PBL in the WRF model, as well as the corresponding error for the ERA5 dataset are presented in Figure 4b,c. The corresponding errors for the MYNN and QNSE schemes are not shown because they had significant errors in predicting the path of the cyclone. Among the WRF simulations, the YSU scheme had the lowest error with respect to JTWC, and the track error decreased along the cyclone path as it approached the landfall location; however, the trend of error for ACM2 scheme was increasing, and then decreasing. YSU was previously suggested as the optimum PBL parameterization for Cyclone Gonu [9]. Unlike YSU, MYJ presented an increasing trend of the track error along the cyclone path. The large deviations along the track indicate the necessity for further research focusing on the model dynamics and initialization. Cyclones from the north-west of the Indian Ocean in their paths to the Arabian Sea usually deviate toward north and make landfall over the coastal regions of India and Pakistan or move toward the coast of the Oman. They rarely penetrate into the Gulf of Oman; therefore, the usual physics implemented in the WRF model might not be able to reproduce an unusual track such as that of Cyclone Shaheen [79–82]. The YSU scheme could successfully simulate the track of five severe cyclonic storms in the Bay of Bengal [39]. Singh and Bhaskaran [24] evaluated the performance of the WRF model in the simulation of tropical cyclone landfall in the Bay of Bengal by five schemes for PBL. They concluded that the YSU scheme led to the most accurate cyclone track. The track of three severe cyclones, Gonu, Phet and Ashobaa, in the Arabian Sea were successfully simulated by using the YSU scheme [9,10,25].

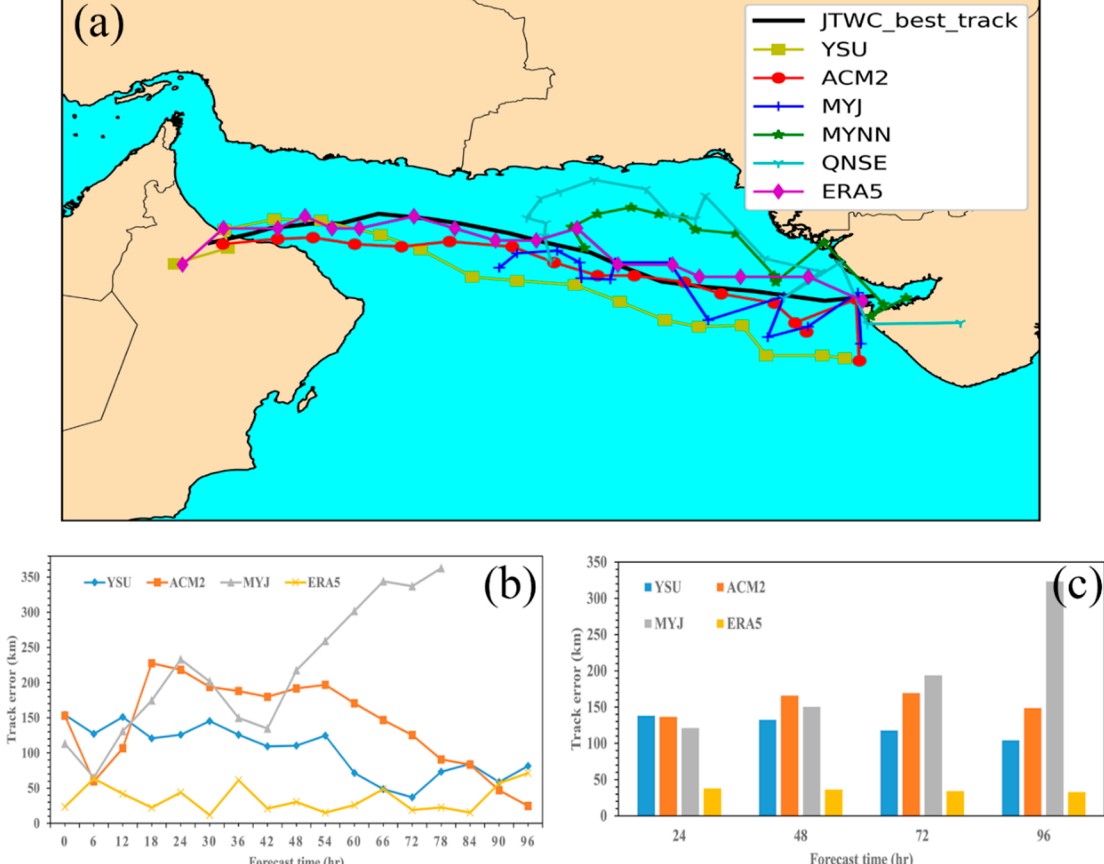

**Figure 4.** Predicted tracks of Cyclone Shaheen with different PBL schemes along with JTWC best-fit track of Cyclone Shaheen, (**a**) track, (**b**,**c**) temporal variation in track error.

The simulated cyclone intensity (hPa) and strength (m/s) by WRF using five parameterizations for PBL and ERA5 reanalysis data were compared against JTWC data in Figure 5. For the first 24 h (i.e., 0000 UTC on 30 September to 0000 UTC on 1 October), the intensity and strength variations by all schemes were in a good agreement with JTWC, except YSU. At this time, the system intensified due to a deep depression from a severe cyclonic storm. It can be noted that the pressure drop (which means intensity) was overestimated with YSU and underestimated with ACM2, MYJ, MYNN, and QNSE (see Figure 5a), while the maximum wind speed was underestimated by the MYJ, MYNN, and QNSE schemes and overestimated by the YSU and ACM2 schemes (see Figure 5b). The YSU scheme resulted in the highest intensification of storm with the minimum sea level pressure of 959.1 hPa and the maximum wind speed of 48.7 m/s. The ACM2 scheme showed two peaks in intensification, one at 0000 UTC on 2 September, and the second one at 1200 UTC on 3 October, with a discontinuous weakening phase between them. It is clear that the intensification and strengthening trend, as well as the cyclone dissipation were well simulated by the ACM2 scheme. The results from Figure 5 clearly indicate that ACM2 outperformed the other options in reproducing the intensification and strengthening trends during the storm life. Note that ERA5 data showed a much weaker storm compared to the intensity of the real storm.

The mean track error, *MBE*, and *RMSE* scores for intensity and strength obtained by the YSU, ACM2, MYJ parameterizations and ERA5 from 30 October to 4 September 2021 are presented in Table 2. The track errors are 104.20, 148.70, 323.09, and 32.65 km for the YSU, ACM2, MYJ schemes, and ERA5, respectively. As expected, due to the nature of reanalysis data, ERA5 has lower error than the simulations do using WRF in reproducing Cyclone Shaheen's track with respect to JTWC observations. Additionally, the ACM2 scheme revealed the lowest intensity and strength errors (intensity error: *MBE* = 0.36 hPa

and *RMSE* = 3.85 hPa; strength error: *MBE* = 2.94 m/s and *RMSE* = 5.09 m/s). MYJ outperformed YSU in reproducing the cyclone strength (MYJ: *MBE* = −4.80 m/s and *RMSE* = 7.98 m/s; YSU: *MBE* = 9.83 m/s and *RMSE* = 10.92 m/s), while the *MBE* and *RMSE* scores for intensity were similar for MYJ and YSU. Although ERA5 presented the highest accuracy for the cyclone track, its significant errors in representing the intensity and strength of the cyclone demonstrated the deficiency of the reanalysis data in reproducing the cyclone eye, as discussed in previous sections.

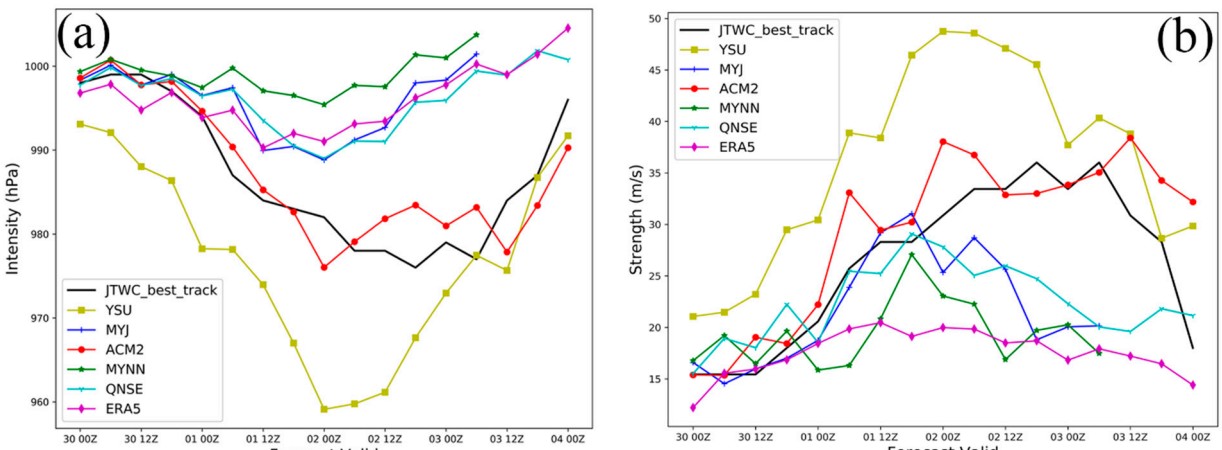

**Figure 5.** Time series of (**a**) intensity (hPa) and (**b**) strength (m/s) of Cyclone Shaheen by five PBL schemes, ERA5, and JTWC data.

**Table 2.** Mean track error, *MBE*, and *RMSE* scores for intensity and strength with respect to three WRF experiments and ERA5. Boldface denotes best score among PBL schemes.

| Experiments | Mean Track Error (km) | Intensity (hPa) | | Strength (m/s) | |
|---|---|---|---|---|---|
| | | *MBE* | *RMSE* | *MBE* | *RMSE* |
| YSU | **104.20** | −9.93 | 11.69 | 9.83 | 10.92 |
| ACM2 | 148.70 | **0.36** | **3.85** | **2.94** | **5.09** |
| MYJ | 323.09 | 9.63 | 12.31 | −4.80 | 7.98 |
| ERA5 | 32.65 | 9.17 | 12.263 | −8.78 | 10.75 |

The simulated wind speed (m/s) and sea level pressure (hPa) by the YSU and ACM2 schemes were compared with available in situ data and ERA5 analysis at Chabahar and Muscat International Airports and the Chabahar buoy (Figure 6). Both the YSU and ACM2 schemes appropriately estimated the speed increase and the pressure drop in the stations during the passage of Cyclone Shaheen. The YSU scheme overestimated the wind speed at Muscat and Chabahar buoy, especially close to the peak of the storm. Except for the Muscat station (located near the storm landfall), the trend of simulated wind speed and pressure from YSU at both Chabahar stations were in a better agreement with the observations and ERA5 data. The path of Cyclone Shaheen in Figure 4a indicated that the Muscat synoptic station was directly exposed to the cyclone during landfall. The wind speed and sea level pressure at this station clearly indicate the weakness of the reanalysis data in estimating the storm intensity and strength. Note that the ACM2 scheme predicted the landfall to occur with, approximately, a 12 h time lag.

Variations in *MBE* versus *RMSE* and *d* for sea level pressure and wind speed at Chabahar buoy and airport stations (Chabahar and Muscat) as a result of the ACM2 and YSU schemes and ERA5 are presented in Figure 7. Although both ACM2 and YSU resulted in identical *d* (~0.6) for sea level pressure at Chabahar buoy, ACM2 outperformed YSU in the *MBE* and *RMSE* scores. The highest *d* and lowest *RMSE* scores were obtained by ERA5. ACM2 remedied the severe underestimation of sea level pressure by YSU and ERA5. Large

*d* and small *RMSE* values were observed by YSU and ERA5 at Chabahar International Airport, though the substantial overestimation of sea level pressure decreased by adopting YSU. Neither the WRF model nor the reanalysis data were successful at simulating the conditions at Muscat International Airport (see Figure 7a).

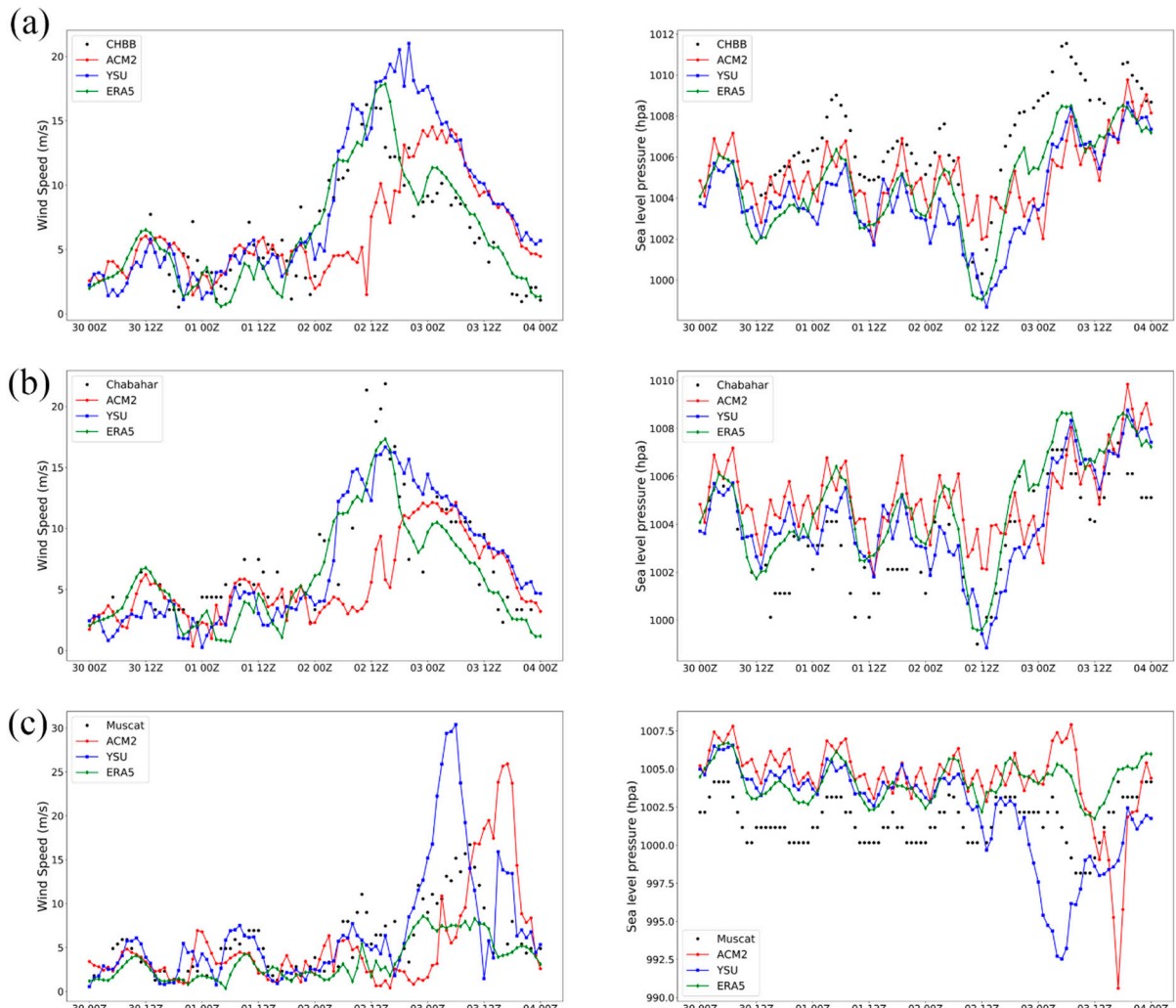

**Figure 6.** Simulated wind speed ($\frac{m}{s}$) and sea level pressure (hPa) against observed and ERA5 data at (**a**) Chabahar buoy, (**b**) Chabahar International Airport, and (**c**) Muscat International Airport.

Similar to the sea level pressure at Chabahar buoy, ACM2 outperformed YSU in reproducing measured wind speed with lower *MBE* and *RMSE* scores, while YSU resulted in higher *d* than that obtained by ACM2. The most accurate wind speed estimation was achieved by ERA5. Although slightly higher/lower *d/RMSE* values were obtained by ERA5 relative to those of YSU at Chabahar International Airport, YSU alleviated the severe underestimation of wind speed by ERA5. The weakest performance in wind speed simulation at Muscat International Airport relates to ACM2. YSU and ERA5 exhibited identical *d*, while the lowest *RMSE* was obtained by ERA5, and the lowest absolute *MBE* was related to YSU (see Figure 7b).

The wind speed along the altimeter tracks provided a great opportunity to assess the spatial quality of wind speed from ERA5 reanalysis wind data, as well as from the WRF model during Cyclone Shaheen. The ERA5 wind data and the simulated wind fields by WRF were compared against the AODN altimeter data, as presented in Figure 8. The ERA5 wind data are well scattered around the perfect fit line (45° line) in Figure 8a. The lowest *MBE* (~0.43 m/s) and *RMSE* (~2.15 m/s) and highest *d* (~0.9) were obtained by

the ERA5 wind data. This could be attributed to data assimilation with satellite data incorporated in the ERA5 dataset. In contrast, the weakest performance in reproducing altimeter wind data was related to the WRF model using YSU for PBL parametrization (see Figure 8b). Additionally, YSU led to the highest *MBE* (~2.59 m/s), indicating severe wind speed overestimation. Unlike YSU, the ACM2 formulation had less bias against the altimeter wind speed, as shown in Figure 8c, with *MBE* = 0.95 m/s.

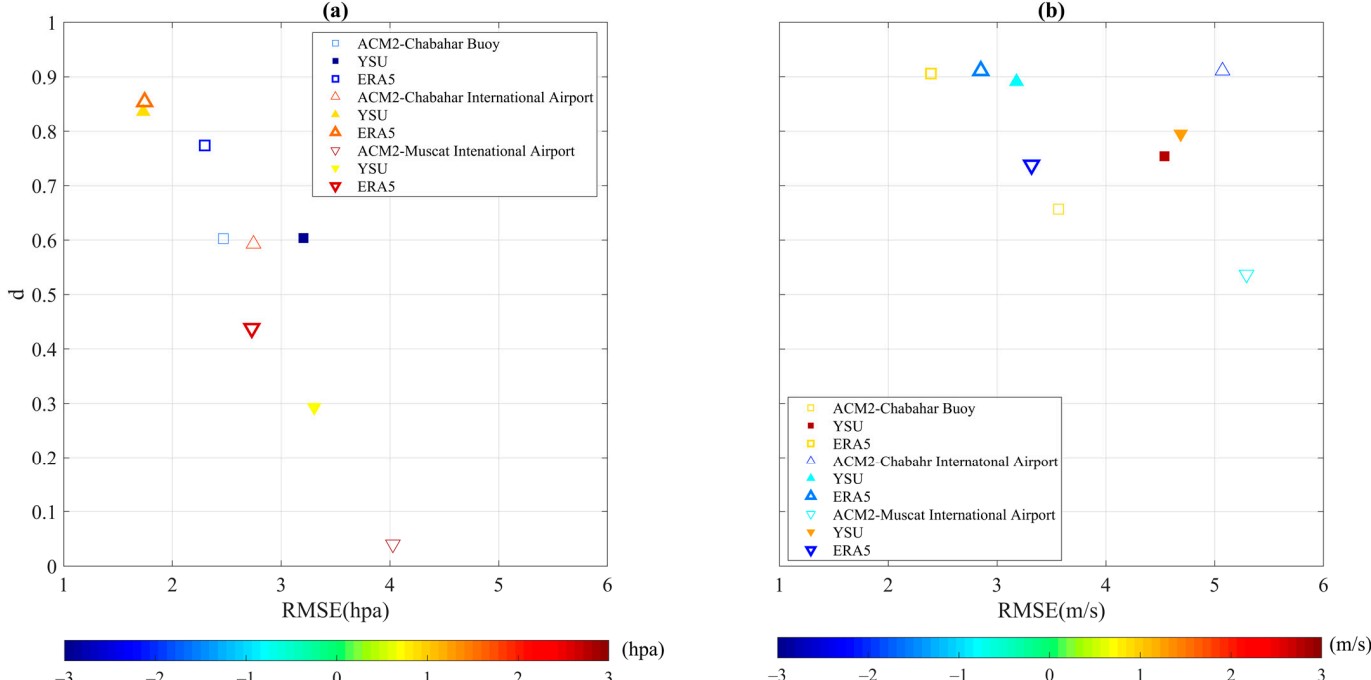

**Figure 7.** Estimated *MBE*, *RMSE*, and *d* for (**a**) sea level pressure and (**b**) wind speed at Chabahar buoy (square), Chabahar International Airport (upward triangles), and Muscat International Airport (downward triangles) by applying ACM2 (plain markers with light edges), YSU (solid markers), and ERA5 (plain parkers with tick edges). Color bars in (**a**,**b**) represents *MBE* score for sea level pressure and wind speed, respectively.

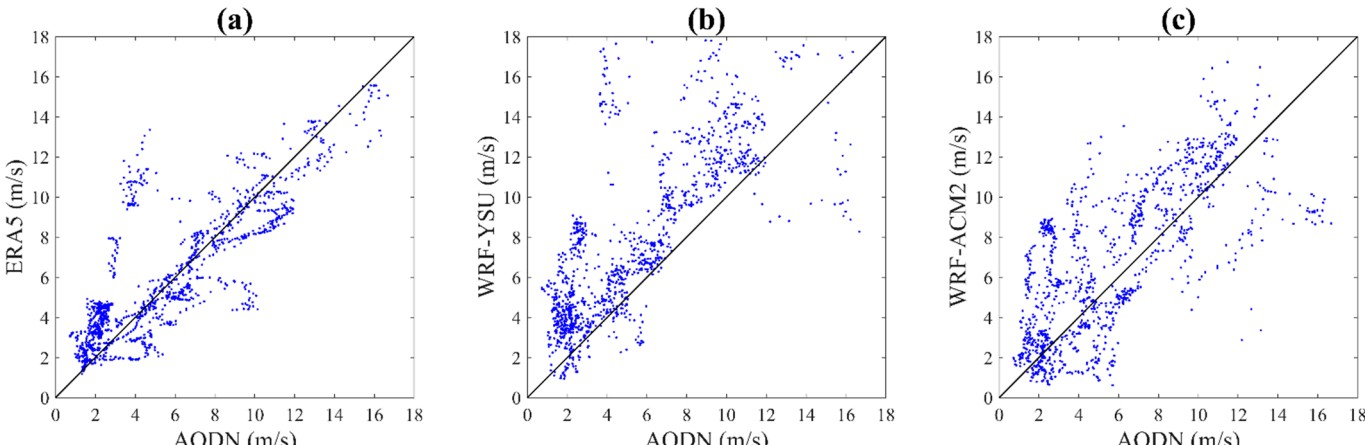

**Figure 8.** The scatter plot of wind speeds (blue dots) from (**a**) ERA5, (**b**) WRF model with YSU, and (**c**) WRF model with ACM2 for PBL vs. AODN recorded wind speed (altimeter) from 30 September to 4 October 2021. The black line shows the 1:1 slope in each case.

Radius–height cross-sections of the azimuthal mean tangential winds at 0000 UTC on 2 October 2021 by the two most successful schemes, YSU and ACM2, are presented in Figure 9. The tangential wind spread from the storm's center to a radius of more than

300 km. The comparison of tangential winds indicates that the lower layer velocities for the YSU and ACM2 schemes are within the ranges of 40–50 and 20–30 m/s, respectively. A relatively narrower and stronger vortex for YSU than that for ACM2 can be seen from the tangential winds distribution. The results of both simulations show that the radius of maximum wind (RMW) extended to ~50 km from the storm center. Storm circulation extends up to pressure level of 200–300 hPa, with the maximum wind speed of ~50 m/s near the 900 hPa pressure level for the YSU scheme, while in ACM2 scheme, it extends up to pressure level of 400–500 hPa. The maximum wind speed of ~30 m/s occurred at a level lower than the 900 hPa pressure level and closer to the surface. Within the boundary layer of the eye region, the tangential winds are weak. Although, the tangential winds sharply increase within RMW, they gradually decrease outside RMW. It is clear that the PBL parametrization has a significant impact on the storm structure.

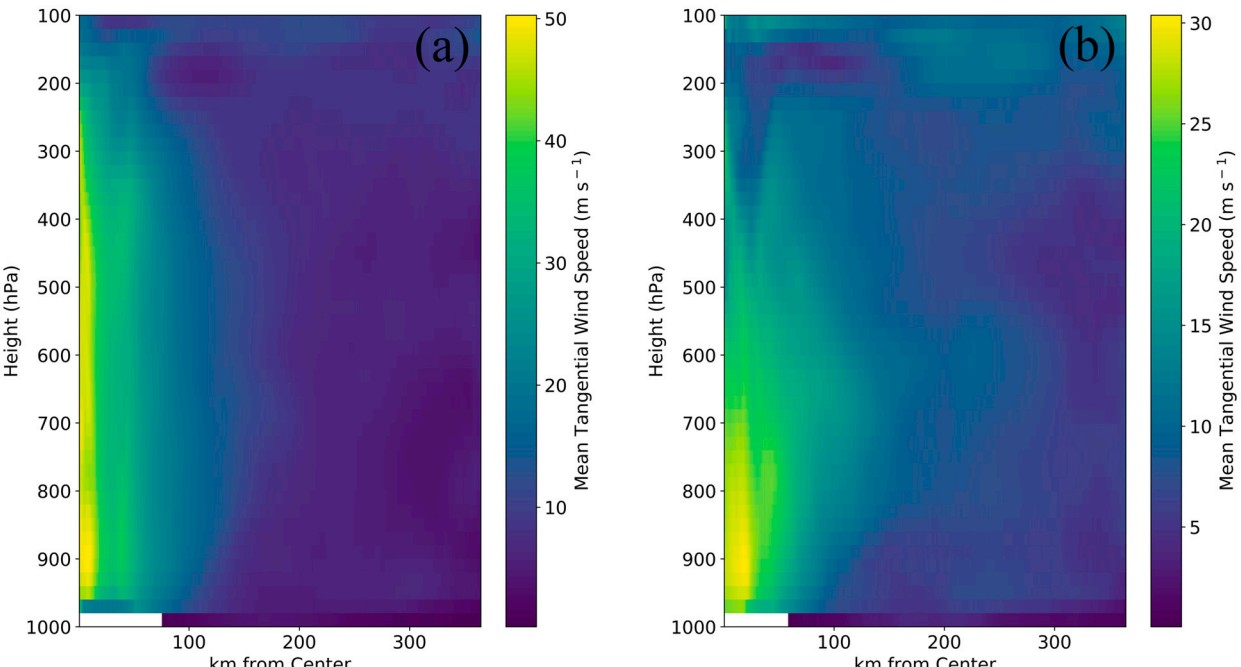

**Figure 9.** Radial–height cross-sections of azimuthally averaged tangential winds at 0000 UTC on 2 October 2021 by (**a**) YSU and (**b**) ACM2 schemes.

### 3.2. Skill Assessment of SWAN Model

The ERA5 wind field was used to force the SWAN model to simulate a cyclone-induced wave in the Arabian Sea. Several whitecapping dissipation formulations, including Komen-Default, Westhuysen-Default, Janssen-Default, and ST6-Default, were examined. The Janssen-BSD2021 and ST6-BSD2021 formulations along with ERA5 wind were two other scenarios that were evaluated. Moreover, hybrid wind data based on ERA5 was applied with Janssen-Default and ST6-Default setups.

The simulated $H_s$, $T_z = T_{m02} = 2\pi \left( \dfrac{\iint \sigma^2 E(\sigma,\theta)d\sigma d\theta}{\iint E(\sigma,\theta)d\sigma d\theta} \right)^{-0.5}$ and mean wave direction from the eight above scenarios were compared with the buoy data at Chabahar from 1200 UTC on 30 September to 0400 UTC on 4 October 2021, as seen in Figure 10. It is clear from Figure 10a that Komen-Default, Westhuysen-Default, Janssen-Default, and ST6-Default in combination with ERA5 wind data successfully reproduced the peak wave height (~3 m) at 1200 UTC on 2 October. The good agreement between the measured and modeled wave heights before and after the peak is obvious, except for ST6-BSD2021, which underestimated the wave height. The peak wave height was overestimated between 1–2 m by the Janssen-Default-Hybrid and ST6-Default-Hybrid scenarios. This could be attributed to higher wind speeds in the hybrid wind field than those in the ERA5 data.

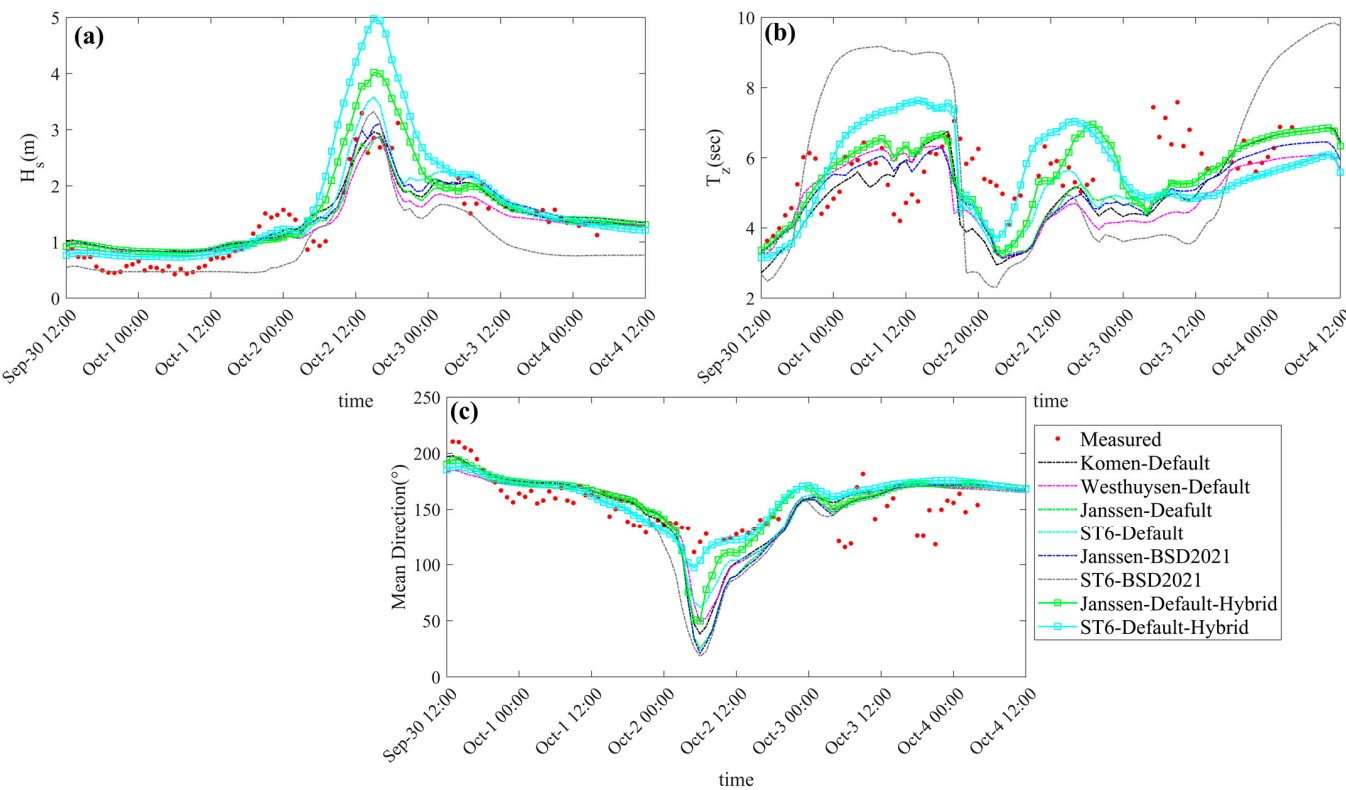

**Figure 10.** Simulated (**a**) $H_s$ (m), (**b**) $T_z$ (s), and (**c**) mean direction (°) compared with Chabahar buoy measured data.

As shown in Figure 10b, the general trend of the measured $T_z$ was well captured by Komen-Default, Westhuysen-Default, Janssen-Default, and ST6-Default. $T_z$ was underestimated from 0000 UTC to 1200 UTC on 2 October using these configurations though. $T_z$ estimated with Janssen-BSD2021 agreed more with the measured data, while ST6-BSD2021 resulted in significant $T_z$ overestimations from 1 to 2 October and 1200 UTC on 3 to 4 October. Lower values for the $T_z$ parameter was obtained from 2 to 1200 UTC on 3 October with ST6-BSD2021. Janssen-Default and ST6-Default using hybrid wind data slightly improved the $T_z$ underestimation from 2 to 3 October. Furthermore, $T_z$ was overestimated from 1 to 2 October by ST6-Default-Hybrid.

The mean wave direction obtained by all the scenarios was compared with the buoy data in Figure 10c. All the scenarios were successful in this regard. Although scenarios with ERA5 depicted a downward pattern from 0000 UTC to 1200 UTC on 2 October, applying the hybrid wind data improved the SWAN model performance.

Statistical indices were employed to evaluate the accuracy of the simulated $H_s$ and $T_z$ against the measured data at Chabahar buoy from 1200 UTC on 30 September to 0400 UTC on 4 October 2021. As shown in Figure 11a, Komen-Default, Westhuysen-Default, Janssen-Default, and ST6-Default using ERA5 resulted in similar *RMSE* (~0.31 m) and *d* (~0.95) values for $H_s$. The lowest *MBE* was obtained by Westhuysen-Default (~0.003 m), while Komen-Default and ST6-Default overestimated the wave height (~0.1 m). ERA5 with Janssen-BSD2021 outperformed ST6-BSD2021 in reproducing the measured $H_s$. The estimated statistical indices for Janssen-BSD2021 were similar to those in the aforementioned scenarios. ST6-BSD2021 severely underestimated the wave height (~−0.3 m) with higher *RMSE* (~0.49 m) and relatively lower *d* values (~0.89). The poorest performance in $H_s$ estimation was observed with using hybrid wind data and ST6-Default. The ST6-Default-Hybrid configuration severely overestimated the wave height (~0.34 m) with the highest *RMSE* (~0.7 m) and the lowest *d* (~0.87); therefore, the best and worst performances were related to Westhuysen-Default with ERA5 and ST6-Default, which applied hybrid wind, respectively.

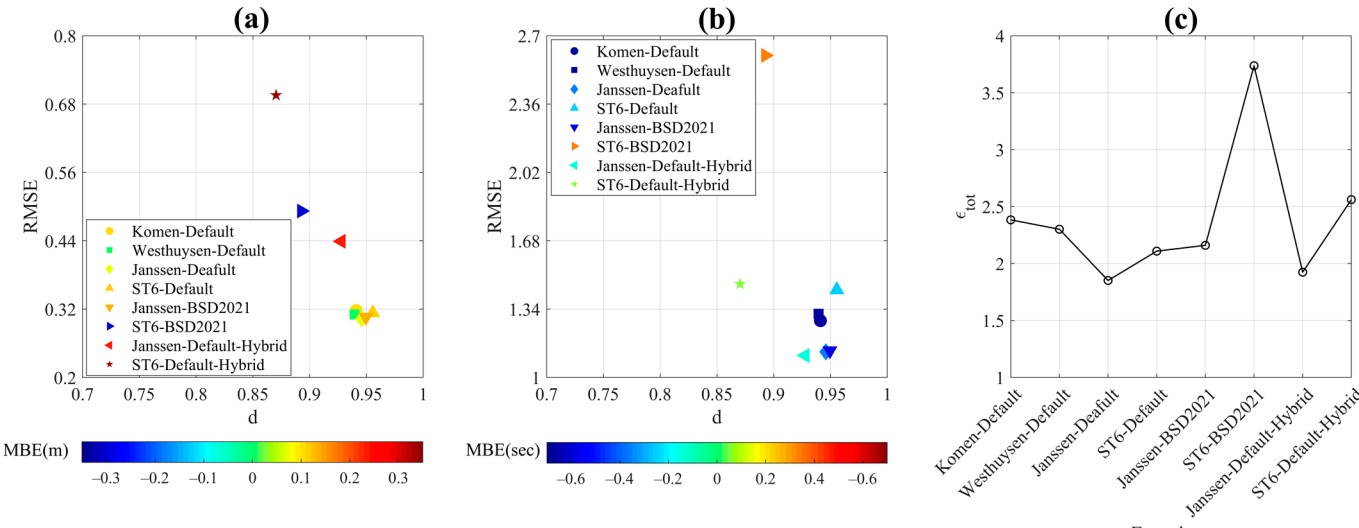

**Figure 11.** Estimated *MBE*, *RMSE*, and *d* indices for (**a**) $H_s$ and (**b**) $T_z$. (**c**) Variations in $\epsilon_{tot}$ (Equation (10)) with different model scenarios.

The values of *RMSE* for the simulated $T_z$ shown in Figure 11b varied in the range of 1.12–1.43 s for Komen-Default, Westhuysen-Default, Janssen-Default, and ST6-Default setups. Komen-Default and Westhuysen-Default severely underestimated $T_z$ (~−0.67 s), while ST6-Default resulted in a relatively less exaggerated underestimation (~−0.24 s). Four formulations with default tuning values in combination with ERA5 data led to similar *d* values (~0.94) for $T_z$. Similar to Komen-Default and Westhuysen-Default, Janssen-BSD2021 with the ERA5 wind data led to a severe underestimation for the $T_z$ parameter (~−0.58 s), while ST6-BSD2021 resulted in an overestimation of ~0.34 s. Janssen-BSD2021 was superior to ST6-BSD2021 in terms of the *RMSE* and *d* indices. In the experiments applying hybrid wind data, the *MBE* index improved considerably. Additionally, ST6-Default-Hybrid performed better than Janssen-Default-Hybrid did. Despite the mentioned superiority of ST6-Default-BSD2021 in *MBE* index, much lower *RMSE* and higher *d* values were presented by the Janssen-Default-Hybrid setup.

Due to the diversity of the SWAN model performances in reproducing $H_s$ and $T_z$ by different formulations for $S_{in}$ and $S_{wc}$ and different wind data, the following error $\epsilon_{tot}$ was introduced to identify the most successful scenarios for $H_s$ and $T_z$ simulation during the dominance of Cyclone Shaheen in the Arabian Sea:

$$\epsilon_{tot} = |MBE|_{H_s} + RMSE_{H_s} + |MBE|_{T_z} + RMSE_{T_z} \qquad (10)$$

Equation (10) was applied by the authors of [78] to select the best configuration of WRF to be used in the WAVEWATCH-III model for wave prediction during the dominance of a meteotsunami in the Persian Gulf. Variations in $\epsilon_{tot}$ with the eight experiments mentioned above are shown in Figure 11c. It is clear that Janssen-Default with ERA5 and the hybrid wind data led to the lowest $\epsilon_{tot}$ values; therefore, these two setups were used for model evaluation of reproducing the cyclone-induced maximum significant wave height.

The cyclone-induced maximum significant wave height from 1 to 4 October 2021 for the Janssen-Default setup by applying ERA5 and the hybrid wind data is shown Figure 12. A higher wave height appears at the right hand side of cyclone track. This can be attributed to higher wind speeds on the right quadrants of cyclones, where the wind vectors were more aligned with the translation speed (rightward bias) [83–86]. The maximum wave height estimated by Janssen-Default-Hybrid (Figure 12b) exceeded 7 m at 1800 UTC on 2 October, while the maximum wave height simulated by Janssen-Default run with ERA5 (Figure 12a) did not exceed 5 m in the first stages of Cyclone Shaheen's passage over the Arabian Sea. As a consequence, the reanalysis ERA5 wind data could not reproduce the

development of Cyclone Shaheen realistically. Similar results were previously reported for Cyclone Gonu in the Arabian Sea [53].

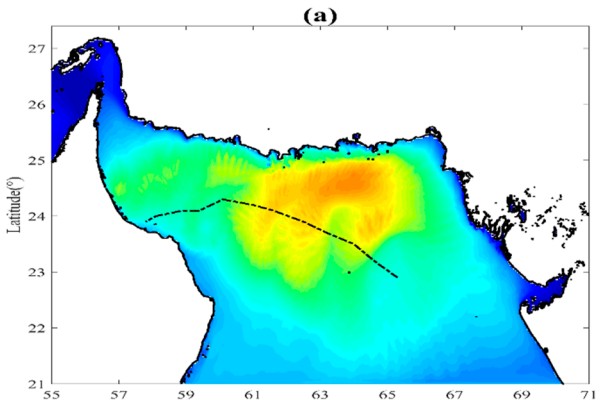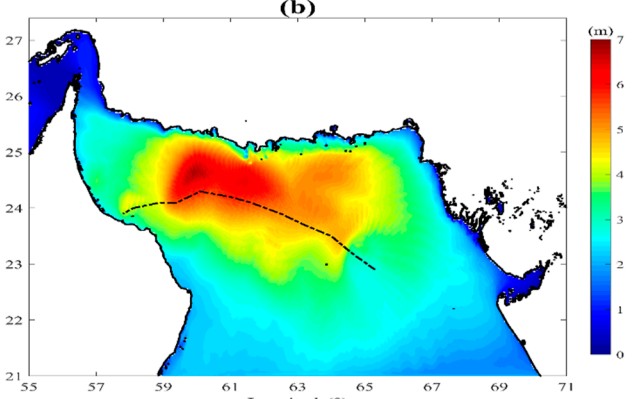

**Figure 12.** Predicted maximum significant wave heights by Janssen-Default using (**a**) ERA5 and (**b**) hybrid wind data. Black dash-dotted lines indicate the track of Cyclone Shaheen extracted from JTWC.

## 4. Summary and Conclusions

This study scrutinized the influence of different PBL schemes incorporated in the WRF model on the prediction of track, intensity, and strength of Cyclone Shaheen in the northern Arabian Sea. The five PBL schemes used for the simulations were YSU, MYJ, MYNN, ACM2, and QNSE. Among these methods, MYJ, MYNN, and QNSE are classified as TKE closure schemes, whereas YSU and ACM2 are based on K-profile formulation. The simulated track, intensity, and strength were evaluated against the JTWC dataset. Moreover, ERA5 reanalysis data, synoptic wind speed observations, and altimeter wind speed data were employed to skill assess the atmospheric model.

The sensitivity analysis of the PBL schemes indicated that the track, intensity, and strength of Cyclone Shaheen were highly dependent on the PBL parametrizations. In general, the local MYJ, MYNN, and QNSE schemes failed to reproduce the cyclone' features appropriately. In contrast, the YSU and ACM2 schemes led to more accurate predictions of the track, intensity, and strength. The lowest mean track error was related to YSU, and then ACM2.

It is noted that prior to the intensification phase, when the system was a tropical storm, the predicted intensity and strength were less sensitive to the PBL schemes. During the intensification stage, there was a large deviation in the simulated intensity and strength predicted using local schemes (MYJ, MYNN, and QNSE). The YSU parametrization resulted in the highest intensification of the storm and the most strengthening. Additionally, the intensification and strengthening trends during the storm life were better simulated by ACM2. Although ERA5 data had the most accurate track, the intensity/strength of Cyclone Shaheen was severely overestimated/underestimated by this dataset.

The comparison of modeled wind speed and pressure simulated by the YSU and ACM2 schemes against data from synoptic and buoy stations proved that these schemes were able to reproduce the increase in wind speed and pressure drop during the storm's passage. The quality of wind speed by ERA5 data and the YSU and ACM2 schemes in the WRF model were assessed using altimeter wind data during the cyclone. The WRF model, by using both YSU and ACM2 schemes, overestimated the altimeter data. The ERA5 data outperformed the WRF model in reproducing altimeter wind speed, most likely due to the data assimilation procedure used in producing the ERA5 dataset.

ERA5 and a hybrid wind field in combination with various formulations for $S_{in}$ and $S_{wc}$ terms in the SWAN model were applied to simulate the wave regime in the Arabian Sea during the presence of Cyclone Shaheen. Buoy data measured at Chabahar station were used to skill assess the accuracy of $H_s$, $T_z$, and mean wave direction simulated by

SWAN. Komen, Westhuysen, Janssen, and ST6 whitecapping formulations with their default coefficients in combination with ERA5 wind field successfully reproduced the peak wave height. The Janssen and ST6 methods in combination with the hybrid wind data overestimated the peak wave height. The same trend was found for $T_z$. The BSD2021 calibration coefficients for the Janssen method were more compatible with the measured data, while the ST6 resulted in intense overestimation of the $T_z$ values. The use of hybrid wind in the SWAN model could not effectively enhance the model performance for $T_z$ prediction. Overall, all the scenarios reproduced the trend of measured mean wave direction with different accuracies. Applying the hybrid wind data improved the wave model in the simulation of $T_z$.

Due to sporadic performances of different scenarios (different formulations and input wind data) in the prediction of the measured $H_s$ and $T_z$, a combined error ($\epsilon_{tot}$) function based on *MBE* and *RMSE* scores for $H_s$ and $T_z$ was employed to determine the optimum scenario. The Janssen formulation with $C_{ds} = 4.5$ and $\delta = 0.5$ using either ERA5 or hybrid wind data resulted in the lowest $\epsilon_{tot}$ (~1.9).

The evaluation of the cyclone-induced maximum significant wave height with the default setup of Janssen and employing the ERA5 and hybrid wind data emphasized the inaccuracies of ERA5 in reproducing the development phase of Cyclone Shaheen. The use of hybrid wind resulted in the realistic evolution of the wave field during the passage of the Cyclone Shaheen over the Arabian Sea.

**Author Contributions:** Conceptualization, M.R.; Formal analysis, M.R. and M.B.; Investigation, M.R.; Methodology, M.B. and S.M.S.; Project administration, S.M.S.; Software, M.R. and M.B.; Supervision, S.M.S.; Validation, M.B.; Writing—original draft, M.R. and M.B.; Writing—review and editing, S.M.S. and M.N.A. All authors have read and agreed to the published version of the manuscript.

**Funding:** This research received no specific grant from any funding agency in the public, commercial, or not-for-profit sectors.

**Institutional Review Board Statement:** Not available.

**Informed Consent Statement:** Not applicable.

**Data Availability Statement:** The simulation results presented in this paper can be regenerated using WRF and SWAN models, and the setups for models were described in the manuscript.

**Conflicts of Interest:** The authors declare that they have no competing interest.

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
