# Peer review of "Simulating Meteorological and Water Wave Characteristics of Cyclone Shaheen"

_atmosphere, doi:10.3390/atmos14030533_

Round 1

Reviewer 1 Report

Review comments on “Simulating Meteorological and Water Wave Characteristics of Cyclone Shaheen”

The authors investigated the sensitivity of the results to the planetary boundary layer (PBL)

parameterization in terms of the track, intensity, strength and structure of the cyclone Shaheen with the high resolution advanced Weather Research and Forecasting (WRF) model. Five experiments are considered with five PBL schemes; Yonsei University (YSU), Mel-

lor–Yamada–Janjić (MYJ), Mellor–Yamada–Nakanishi–Niino level 2.5 (MYNN), Asymmetric Convective Model version 2 (ACM2), Quasi-Normal Scale Elimination (QNSE). The track, intensity and strength of experiments are compared with Joint Typhoon Warning Centre (JTWC) dataset. Results have implied the high dependency of the track, intensity and strength of the Cyclone to the PBL parameterization. Simulated tracks with non-local PBL schemes (YSU and ACM2) outperformed the local PBL schemes (MYJ, MYNN and QNSE), especially during the rapid intensification phase of the Shaheen Cyclone before landfall. The results are very encouraging; however, some improvements are needed according to the following comments and suggestions:

Major comments:

  1. In Fig. 2, the time and the image are not consistent.

  2. In Fig. 4a, ACM2 is better than YSU at most of the time; however it is not shown in Fig. 4b, what causes that? Why is the cyclone starting point different for different experiments?

  3. In Figure 5, ACM2 has the best intensity and strength.

  4. What’s the statistical results for Fig. 6? 

  5. In Figure 7, x-axis label is not correct.

  6. There are too many “Error! Reference source not found.”  in the paper.

Minor comments:

  1. Hs and Tz are not defined in abstract.

Author Response

Dear Editor

Thank you for providing us the opportunity to improve this manuscript. The comments from

reviewer is addressed as presented here.

Reviewer1

The authors investigated the sensitivity of the results to the planetary boundary layer (PBL) parameterization in terms of the track, intensity, strength and structure of the cyclone Shaheen with the high resolution advanced Weather Research and Forecasting (WRF) model. Five experiments are considered with five PBL schemes; Yonsei University (YSU), Mellor–Yamada–Janjić (MYJ), Mellor–Yamada–Nakanishi–Niino level 2.5 (MYNN), Asymmetric Convective Model version 2 (ACM2), Quasi-Normal Scale Elimination (QNSE). The track, intensity and strength of experiments are compared with Joint Typhoon Warning Centre (JTWC) dataset. Results have implied the high dependency of the track, intensity and strength of the Cyclone to the PBL parameterization. Simulated tracks with non-local PBL schemes (YSU and ACM2) outperformed the local PBL schemes (MYJ, MYNN and QNSE), especially during the rapid intensification phase of the Shaheen Cyclone before landfall. The results are very encouraging; however, some improvements are needed according to the following comments and suggestions:

Major comments:

  1. In Fig. 2, the time and the image are not consistent.

The time and images are arranged as follows: 0500 UTC (a) on 30 October, (b) on 1 September and (c) on 3 September 2021. The arrangement of (a)-(c) in Figure 2 is right to left to depict the cyclone entrance, evolution and passage over the Arabian Sea.

  1. In Fig. 4a, ACM2 is better than YSU at most of the time; however, it is not shown in Fig. 4b, what causes that? Why is the cyclone starting point different for different experiments?

Although track by ACM2 apparently is closest one to the track of JTWC in Figure 4, lower temporal lag by YSU (-6 hour by YSU against +12 hour by ACM2) was led to lower track error and more accurate simulated track. All tracks in Figure 4a were presented from 0000 UTC on 30 September to 0000 UTC on 4 October 2021. Due to compatibility between tracks by five adopted PBL schemes, ERA5 and the best track by JTWC, all tracks in Figure 4a were provided at 0000, 0600, 1200, 1800 UTC. It is expected that different PBL schemes resulted in different cyclone trajectory even at starting point and landfall.

  1. In Figure 5, ACM2 has the best intensity and strength.

YSU scheme resulted in the highest intensification of storm with a sea level pressure of 959.1 hPa and the highest strengthening with maximum winds of 48.7 m/s. The ACM2 scheme showed two peaks in intensification, one at 0000 UTC, 2 September and the second one at 1200 UTC, 3 October, with a discontinuous weakening phase between them. It is clear that the intensification and strengthening trend as well the cyclone dissipation was well simulated by the ACM2 scheme. The results from Figure 5 clearly indicate that ACM2 outperformed other option in reproducing the intensification and strengthening trend in the duration of storm life.

  1. What’s the statistical results for Fig. 6?

Added to the text

  1. In Figure 7, x-axis label is not correct.

Corrected.

  1. There are too many “Error! Reference source not found.”  in the paper

Corrected.

Minor comments:

  1.  and  are not defined in abstract.

significant wave height () and zero up-crossing wave period () were added in abstract.

Reviewer 2 Report

Authors presented the manuscript entitled “Simulating Meteorological and Water Wave Characteristics of Cyclone Shaheen”. After analyzing this manuscript, we concluded that the study was carried out at a good scientific level, and the results may be of interest to specialists. However, we have several important questions. We recommend the authors to consider these issues and improve the manuscript.

Major notices:

- To study the structure of atmospheric flows when cyclone passes, the authors use a well-known powerful tool for simulation – a Weather Research and Forecast-
ing (WRF) model. Authors used 5 schemes for parametrization of atmospheric boundary layer: Yonsei University (YSU), Mellor–Yamada–Janjić (MYJ), Mellor–Yamada–Nakanishi–Niino level 2.5 (MYNN), Asymmetric Convective Model version 2 (ACM2), Quasi-Normal Scale Elimination (QNSE).
The important conclusion has been obtained: YSU or ACM2 parametrization schemes overestimate wind speed compared with measurement data. However, we believe that, first of all, in order to made such conclusion, the authors need clarify the details. What coefficients do you use in each parametrization of the atmospheric boundary layer. For example, please consider YSU schema. What altitude dependencies of the turbulent diffusion coefficient did you use?

- Many studies show that the MRF and YSU schemes are accurate for parametrization of the atmospheric boundary layer. It may be recommended to compare your results obtained for atmospheric conditions with regional features with the results of other studies:

i) Ma, H.; Cao, X.; Ma, X.; Su, H.; Jing, Y.; Zhu, K. Improving the Wind Power Density Forecast in the Middle- and High-Latitude Regions of China by Selecting the Relatively Optimal Planetary Boundary Layer Schemes. Atmosphere 2022, 13, 2034

ii)Shikhovtsev, A.Y.; Kovadlo, P.G.; Lezhenin, A.A.; Korobov, O.A.; Kiselev, A.V.; Russkikh, I.V.; Kolobov, D.Y.; Shikhovtsev, M.Y. Influence of Atmospheric Flow Structure on Optical Turbulence Characteristics. Appl. Sci. 2023, 13, 1282. https://doi.org/10.3390/app13031282

- How did you determine the trajectory of the cyclone. Did you use the cyclone center trajectory? How was the center of the cyclone determined? What are the errors in determining the center of the cyclone. This question is quite important because a real atmospheric vortex can diffuse (destroy), several centers can appear or Cyclone may deepen. How did you solve these issues and what stages of the life of the cyclone did you consider?

- You are using reanalysis ERA-5. How representative are the ERA-5 data within the region you are examining. Have studies related to comparison of ERA-5 data and radiosonde data been carried out?

Minor notices:

- The abstract contains the reanalysis abbreviation “ERA5 ECMWF”. Although the reanalysis ERA5 ECMWF is well known, we ask you to give the full title of this database.

- The document contains a number of errors related to links (Error! Reference source not found.) Please correct.

-Is it possible to also ask the authors to clarify how cloudy the region within the first and second domains? Just add a few sentences, please.

- Introduction. We recommend authors to discuss the possibilities of use of reanalysis data for atmospheric studies. The reanalysis data including ERA-5 may inaccurate atmospheric reconstruction in mountains, over water, or over coastlines. There are a large number of papers devoted to the reanalysis data verification, including not only wind speed components, air temperatures but also water vapor, cloudiness an so on:

iii) Shikhovtsev, A.Y.; Kovadlo, P.G.; Khaikin, V.B.; Kiselev, A.V. Precipitable Water Vapor and Fractional Clear Sky Statistics within the Big Telescope Alt-Azimuthal Region. Remote Sens. 2022, 14, 6221.

iv) Virman, M.; Bister, M.; Räisänen, J.; Sinclair, V.A.; Järvinen, H. Radiosonde comparison of ERA5 and ERA-Interim reanalysis datasets over tropical oceans. Tellus A Dyn. Meteorol. Oceanogr. 2021, 73, 1–7.

Author Response

Dear Editor

Thank you for providing us the opportunity to improve this manuscript. The comments from

reviewer is addressed as presented here.

Reviewer2

Authors presented the manuscript entitled “Simulating Meteorological and Water Wave Characteristics of Cyclone Shaheen”. After analyzing this manuscript, we concluded that the study was carried out at a good scientific level, and the results may be of interest to specialists. However, we have several important questions. We recommend the authors to consider these issues and improve the manuscript.

Major notices:

- To study the structure of atmospheric flows when cyclone passes, the authors use a well-known powerful tool for simulation – a Weather Research and Forecast-
ing (WRF) model. Authors used 5 schemes for parametrization of atmospheric boundary layer: Yonsei University (YSU), Mellor–Yamada–Janjić (MYJ), Mellor–Yamada–Nakanishi–Niino level 2.5 (MYNN), Asymmetric Convective Model version 2 (ACM2), Quasi-Normal Scale Elimination (QNSE). The important conclusion has been obtained: YSU or ACM2 parametrization schemes overestimate wind speed compared with measurement data. However, we believe that, first of all, in order to made such conclusion, the authors need clarify the details. What coefficients do you use in each parametrization of the atmospheric boundary layer. For example, please consider YSU schema. What altitude dependencies of the turbulent diffusion coefficient did you use?

The aim of present study is to skill assess the different schemes for PBL with their default configurations; therefore, the default mode has been used for the turbulence diffusion coefficients in the WRF model. Turbulence and mixing option (Diff_opt), eddy coefficient option (_opt) was to 1 and 4, respectively in our simulations.

- Many studies show that the MRF and YSU schemes are accurate for parametrization of the atmospheric boundary layer. It may be recommended to compare your results obtained for atmospheric conditions with regional features with the results of other studies:

  1. i) Ma, H.; Cao, X.; Ma, X.; Su, H.; Jing, Y.; Zhu, K. Improving the Wind Power Density Forecast in the Middle- and High-Latitude Regions of China by Selecting the Relatively Optimal Planetary Boundary Layer Schemes. Atmosphere 2022, 13, 2034

ii)Shikhovtsev, A.Y.; Kovadlo, P.G.; Lezhenin, A.A.; Korobov, O.A.; Kiselev, A.V.; Russkikh, I.V.; Kolobov, D.Y.; Shikhovtsev, M.Y. Influence of Atmospheric Flow Structure on Optical Turbulence Characteristics. Appl. Sci. 2023, 13, 1282. https://doi.org/10.3390/app13031282

Few more studies were used for comparison as suggested by the reviewer; e.g.

Rajeswari, et al. [1] with using YSU scheme in WRF could successfully simulate the track of five very severe cyclonic storm in the Bay of Bengal. Singh and Bhaskaran [2] evaluated the performance of WRF model for determination of tropical cyclone landfall in the Bay of Bengal by five schemes for PBL. They concluded that YSU scheme led to most accurate cyclone track. The track of three severe cyclones Gonu, Phet and Ashobaa in the Arabian Sea well simulated by using YSU for PBL in the WRF model [3-5].

- How did you determine the trajectory of the cyclone. Did you use the cyclone center trajectory? How was the center of the cyclone determined? What are the errors in determining the center of the cyclone. This question is quite important because a real atmospheric vortex can diffuse (destroy), several centers can appear or Cyclone may deepen. How did you solve these issues and what stages of the life of the cyclone did you consider?

The location of lowest simulated sea level pressure was considered as cyclone track. This simple derivation of track works well for an intense storm; however, this simple approximation begins to break down as the storm weakens and the sea level pressure gradients between the storm and surrounding environment are reduced [6]. It is inferable that the determination of landfall location, which is important to forecast cyclone-induced high waves and storm surges in coastal communities, with using this simple approach is inaccurate. At this stage more advanced methods such as GFDL should be considered.

- You are using reanalysis ERA-5. How representative are the ERA-5 data within the region you are examining. Have studies related to comparison of ERA-5 data and radiosonde data been carried out?

In general cyclone track by reanalysis wind data are sufficiently accurate, while the wind speed and pressure are under- and overestimated respectively. This align with our findings in present research. Although the deficiency of ERA5 in cyclone intensity and strength prediction is obvious, it could be skillful against earth surface measured data. Moreover, ERA5 wind speed is in more agreement with AODN altimeter data. The quality of ERA5 wind speed over the Gulf of Oman and Arabian Sea in 2010, 2013 and 2015 was assessed with using Ifremer altimeter data by Beyramzadeh, et al. [7]. They showed high consistency of ERA5 wind speed and remotely sensed data over the Gulf of Oman and Arabian Sea.

It is noteworthy to mention that no any research was conducted about the comparison of ERA5 and radiosonde data during the dominance of cyclone in the Arabian Sea. The reviewer suggested article ‘Radiosonde comparison of ERA5 and ERA-Interim reanalysis datasets over tropical oceans’ about the comparison of reanalysis and radiosonde data was added in introduction.

Minor notices:

- The abstract contains the reanalysis abbreviation “ERA5 ECMWF”. Although the reanalysis ERA5 ECMWF is well known, we ask you to give the full title of this database.

ERA5 (fifth generation European Centre for Medium-Range Weather Forecasts) was modified in abstract

- The document contains a number of errors related to links (Error! Reference source not found.) Please correct.

Corrected

-Is it possible to also ask the authors to clarify how cloudy the region within the first and second domains? Just add a few sentences, please.

Below paragraph was added in Manuscript:

At 0000 UTC on 30 September 2021, scattered to broken low and medium clouds with embedded intense to very intense convection lay have formed over Northeast and adjoining East-Central Arabian Sea. Afterward, the system as a severe cyclonic storm over Northeast Arabian Sea moved West-Northwestwards associated broken low and medium clouds with embedded intense to very intense convection lay. In landfall stage the system moved west-southwestwards and weaken into a deep depression, associated broken low/medium clouds with embedded moderate to intense convection over the Oman and Gulf of Oman. 

- Introduction. We recommend authors to discuss the possibilities of use of reanalysis data for atmospheric studies. The reanalysis data including ERA-5 may inaccurate atmospheric reconstruction in mountains, over water, or over coastlines. There are a large number of papers devoted to the reanalysis data verification, including not only wind speed components, air temperatures but also water vapor, cloudiness and so on:

iii) Shikhovtsev, A.Y.; Kovadlo, P.G.; Khaikin, V.B.; Kiselev, A.V. Precipitable Water Vapor and Fractional Clear Sky Statistics within the Big Telescope Alt-Azimuthal Region. Remote Sens. 2022, 14, 6221.

  1. iv) Virman, M.; Bister, M.; Räisänen, J.; Sinclair, V.A.; Järvinen, H. Radiosonde comparison of ERA5 and ERA-Interim reanalysis datasets over tropical oceans. Tellus A Dyn. Meteorol. Oceanogr. 2021, 73, 1–7.

Below paragraph was added in introduction:

Not only wind field, but also some imperfection in humidity, temperature, precipitation and evaporation by reanalysis data was mentioned in many previous studies. Virman, et al. [8] compared ERA5 and ERA-Interim against radiosonde observations over tropical oceans. They concluded that temperature and relative humidity by ERA5 and ERA-Interim deviate from observations at many atmospheric levels; however, ERA-Interim in more agreement than ERA5 from low- to mid-troposphere. Jiao, et al. [9] evaluated the temporal and spatial performance of ERA5 precipitation from 1979 to 2018 against observation data in China. Although the ERA5 reproduced temporal and spatial patterns of measured data with high accuracy, precipitation overestimated in Chinese mainland during summer. The low correlation (~0.57) of precipitation water vapor by ERA5 and radiometer data in short assessment was observed, while for longer duration the consistency of precipitation water vapor by ERA5 with radiometric measured data was improved [10]

  1. Rajeswari, J.; Srinivas, C.; Mohan, P.R.; Venkatraman, B. Impact of boundary layer physics on tropical cyclone simulations in the Bay of Bengal using the WRF model. Pure and Applied Geophysics 2020, 177, 5523-5550.
  2. Singh, K.; Bhaskaran, P.K. Impact of PBL and convection parameterization schemes for prediction of severe land-falling Bay of Bengal cyclones using WRF-ARW model. Journal of Atmospheric and Solar-Terrestrial Physics 2017, 165, 10-24.
  3. Alimohammadi, M.; Malakooti, H. Sensitivity of simulated cyclone Gonu intensity and track to variety of parameterizations: Advanced hurricane WRF model application. Journal of Earth System Science 2018, 127, 1-15.
  4. Alimohammadi, M.; Malakooti, H.; Rahbani, M. Comparison of momentum roughness lengths of the WRF-SWAN online coupling and WRF model in simulation of tropical cyclone Gonu. Ocean Dynamics 2020, 70, 1531-1545.
  5. Soltanpour, M.; Ranji, Z.; Shibayama, T.; Ghader, S. Tropical Cyclones in the Arabian Sea: overview and simulation of winds and storm-induced waves. Natural Hazards 2021, 1-22.
  6. Zambon, J.B.; He, R.; Warner, J.C.; Hegermiller, C.A. Impact of SST and surface waves on Hurricane Florence (2018): A coupled modeling investigation. Weather and Forecasting 2021, 36, 1713-1734.
  7. Beyramzadeh, M.; Siadatmousavi, S.M.; Derkani, M.H. Calibration and skill assessment of two input and dissipation parameterizations in WAVEWATCH-III model forced with ERA5 winds with application to Persian Gulf and Gulf of Oman. Ocean Engineering 2021, 219, 108445.
  8. Virman, M.; Bister, M.; Räisänen, J.; Sinclair, V.A.; Järvinen, H. Radiosonde comparison of ERA5 and ERA-Interim reanalysis datasets over tropical oceans. Tellus A: Dynamic Meteorology and Oceanography 2021, 73, 1-7.
  9. Jiao, D.; Xu, N.; Yang, F.; Xu, K. Evaluation of spatial-temporal variation performance of ERA5 precipitation data in China. Scientific Reports 2021, 11, 1-13.
  10. Shikhovtsev, A.Y.; Kovadlo, P.G.; Khaikin, V.B.; Kiselev, A.V. Precipitable Water Vapor and Fractional Clear Sky Statistics within the Big Telescope Alt-Azimuthal Region. Remote Sensing 2022, 14, 6221.

Round 2

Reviewer 1 Report

The authors' replies solve my concern; in my point of view, the paper can be accepted for publishing.

Reviewer 2 Report

I found that submited manuscript has been improved. I recommend this study for publication.